

# Mass testing of SiPMs for the CMVD at IICHEP

Mamta Jangra[1,2]⋆, Raj Bhupen[1,2], Gobinda Majumder[2], Kiran Gothe[2], Mandar Saraf[2], Nandkishor Parmar[2], Bheesette Satyanarayana[2], Ravindra R. Shinde[2], Shobha K. Rao[2], Suresh S. Upadhya[2], Vivek M. Datar[3], Douglas A. Glenzinski[4], Alan Bross[4], Anna Pla-Dalmau[4], Vishnu V. Zutshi[5], Robert Craig Group[6] and Edmond Craig Dukes[6]

**1** Homi Bhabha National Institute, Mumbai-400094, India
**2** Tata Institute of Fundamental Research, Mumbai-400005, India
**3** The Institute of Mathematical Sciences, Chennai-600113, India
**4** Fermi National Accelerator Laboratory, IL 60510, United States
**5** Northern Illinois University, IL 60510, United States
**6** Virginia University, VA, United States

⋆ mamtajangra894@gmail.com

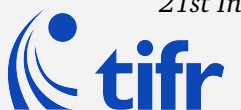

## Abstract

A Cosmic Muon Veto Detector (CMVD) is being built around the mini-Iron Calorimeter (mini-ICAL) detector at the transit campus of the India based Neutrino Observatory, Madurai. The CMV detector will be made using extruded plastic scintillators with embedded wavelength shifting (WLS) fibres which propagate re-emitted photons of longer wavelengths to silicon photo-multipliers (SiPMs). The SiPMs detect these scintillation photons, producing electronic signals. The design goal for the cosmic muon veto efficiency of the CMV is >99.99% and fake veto rate less than $10^{-5}$. A testing system was developed, using an LED driver, to measure the noise rate and gain of each SiPM, and thus determine its overvoltage ($V_{ov}$). This paper describes the test results and the analysed characteristics of about 3.5k SiPMs.



## 1   Introduction

The Iron-CALorimeter (ICAL) detector is the proposed detector by India based Neutrino Observatory (INO) to study the properties of atmospheric neutrinos. It is planned to build under a rock cover of $\sim$1.2 km to reject the cosmic muon background. A depth of $\sim$1.2 km reduces the cosmic muon flux by an order of $10^6$ and a shallow depth of $\sim$100 m reduces the cosmic muon flux by an order of $10^2$.

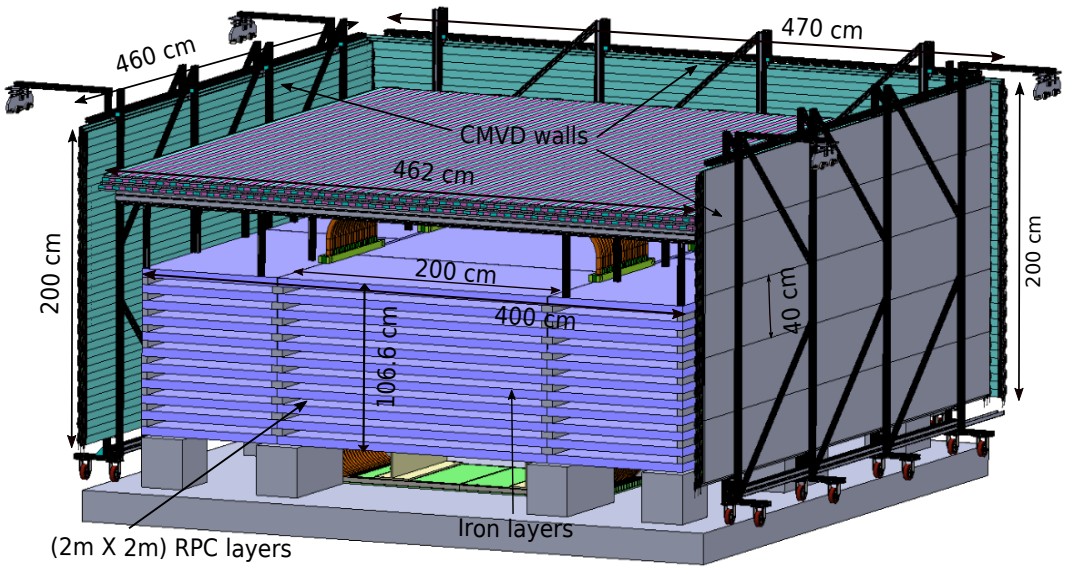

Figure 1: A schematic of the Cosmic Muon Veto Detector around mini-ICAL.

The mini-ICAL detector i.e. $(1/600)^{th}$ scaled down version of ICAL detector, is currently operational at Madurai, India. The mini-ICAL detector has 20 (2 m×2 m) Resistive Plate Chambers (RPCs) sandwiched between 11 layers of iron, 10 RPCs on the front and 10 on the back side. To check the feasibility of a shallow depth neutrino experiment, a CMVD is being built around the mini-ICAL detector. A schematic of the CMVD around the mini-ICAL is shown in Fig. 1. The installation of the CMVD will require a total of 736 extruded plastic scintillators and 2944 SiPMs. It is mandatory to test all the components of the CMVD before the installation. A total of 3488 SiPMs were tested as a part of R&D for the CMVD.

## 2 About Silicon photomultiplier

A silicon photomultiplier is a solid state photosensor which is an array of microcells. Each microcell is an avalanche photo-diode and are connected in parallel to each other with a common bias. Each microcell has a quenching resistor in series to quench the avalanche.

The SiPM model S13360-2050VE from Hamamatsu is used for the CMVD experiment. This particular model of SiPM has a total of 1584 microcells, an effective photosensitive area of 2 mm×2 mm, microcell pitch of 50 $\mu$m, fill factor of 74%, the breakdown voltage ($V_{br}$) of (53 ± 5) V at room temperature [2]. The WLS fibre of diameter 1.4 mm couples perfectly with 2 mm×2 mm SiPM. The SiPMs are mounted on a panel as shown in Fig. 2 on the right and each panel has 16 SiPMs. The detailed view of a single SiPM is shown in the Fig. 2 on the left.

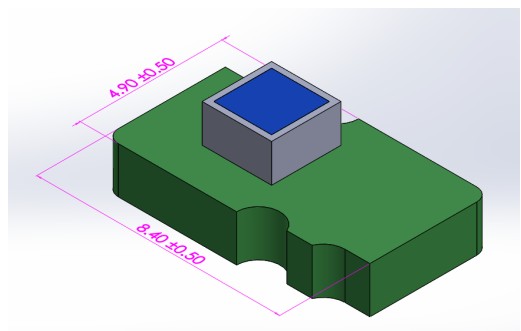
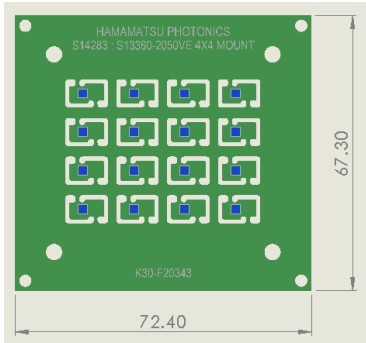

Figure 2: SiPM on a tiny carrier board (on left), a panel with 16 SiPMs (on right).

## 3 Experimental setup

The testing is performed in a lightproof black box. The SiPM panel is mounted on one of the face of the black box as shown in the Fig. 3. An LED system (CAEN SP5601) is used for the SiPM testing purpose. The LED system has an ultrafast LED driver which is used to expose the light on the SiPM panel. Along with the LED driver, the LED system has an external pulser with a trigger synchronized with the LED pulse. The LED driver can send a bunch of few photons upto tens of photons on every trigger.

A piece of tyvek paper is used to diffuse the light uniformaly on all the SiPMs. The raw signals from all 16 SiPMs are amplified using transimpedance (TI) amplifiers [3] and the amplified signals are connected to Domino Ring Sampler (DRS) boards. A total of four DRS boards are used for this testing. For the data collection, the trigger from the external pulser is sent to one of the DRS boards and it was synchronized with other DRS boards in a daisychain mode. The DRS boards are connected to a computer system to collect the data. A schematic of the electronic circuit for signal readout is shown in Fig. 4.

The charge generated by the SiPM is measured from the raw signals using the equation 1

$$q = \frac{1}{R \times G} \int_{to}^{t1} V(t) dt,$$ (1)

where R is the input resistance of the TI amplifier and $G$ is the gain of the operational amplifier stage. A total of 218 SiPM panels were tested using this setup.



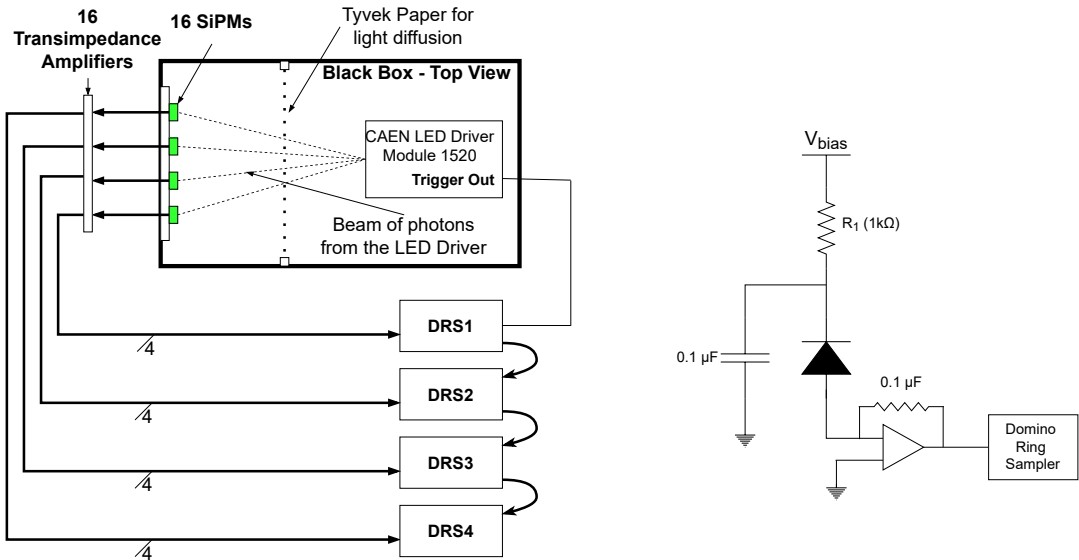

Figure 3: The SiPM mass testing experimental setup.    Figure 4: SiPM circuit diagram.

## 4 Test results

A typical example of SiPM output signal in response to LED excitation is shown in Fig. 5a. For each SiPM, a total of 5000 events are collected. The signal is inetegrated within a 100 ns window for charge collection and the pedestal is subtracted to correct for the baseline fluctuations. The corresponding charge distribution with seceral photoelectron (pe) peaks is shown in Fig. 5b for one of the SiPMs at $V_{bias} = 54$ V.

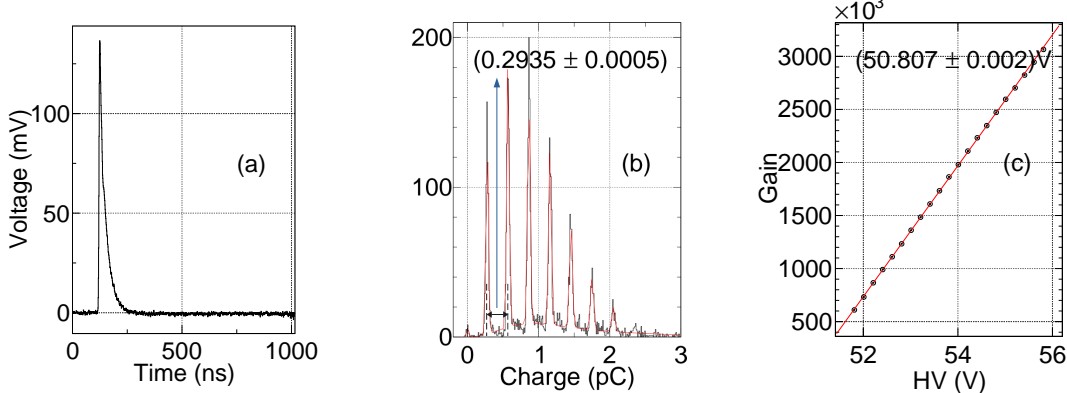

Figure 5: (a) Raw SiPM signal, (b) Charge distribution at $V_{bias} = 54$ V and (c) Calibration plot for measuring $V_{br}$ and dG/dV for one of the SiPMs.

The distribution is fitted with the function:

$$f(y) = Landau(y) + \sum_{n=0}^{N-1} A_n \times e^{-\frac{(y-n\mu)^2}{2n\sigma^2}},$$ (2)

where N is the number of photoelectron (p.e.) peaks, $A_n$ is the peak height, $\mu$ is the average gap between the consecutive photoelectron peaks and $\sigma$ is the gaussian width of p.e. peak. The average gap between the consecutive p.e. peaks ($\mu$) is measured from the fit and gain is

calculated as follows:

$$Gain(G) = \frac{\mu}{(1.6 \times 10^{-19})C},$$

where $(1.6 \times 10^{-19})$C is the elementary charge. The data is collected for five different values of bias voltage ($V_{bias}$). The testing with LED system as well as the noise data collection are done at room temperature (25 °C) and the temperature was recorded. The gain versus $V_{bias}$ is plotted and fitted with a linear function as shown in Fig. 5c. From the linear fit, the slope is a measure of change in gain with respect to change in bias voltage i.e. dG/dV and the ratio of intercept to slope is a measure of $V_{br}$ and is estimated for each of these SiPMs.

## 5 Summarised results

Fig. 6 shows the dG/dV and $V_{br}$ values for all 3488 SiPMs. A total of three bands are observed for dG/dV and $V_{br}$ measurements corresponding to the different SiPMs. The observed $V_{br}$ values are consistent with the $V_{br}$ values given by Hamamatsu, at 25 °C. The majority of the SiPMs are having dG/dV values $\sim 6.8 \times 10^5$. It is observed that one of SiPMs has low gain i.e. $5.1 \times 10^5$ and one has a physical damage. It is clear that all except two SiPMs (one with low gain and the other with physical damage) are suitable for the CMVD purpose.

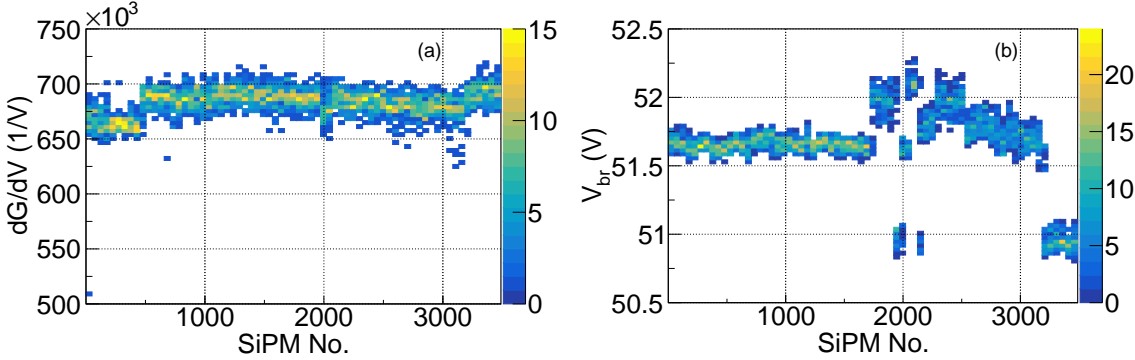

Figure 6: (a) The variation in gain with respect to bias voltgae (dG/dV) versus SiPM number and (b) $V_{br}$ versus SiPM number at room temperature (25 °C).

Another important parameter is the noise rate of the SiPM. Thus, along with LED calibration, noise data is collected using a random trigger at $V_{bias} = 54$ V. Fig. 7 shows the noise rate of SiPMs at a threshold of 0.5 pe. The noise rates are within the tolerable range as per previous studies [1].

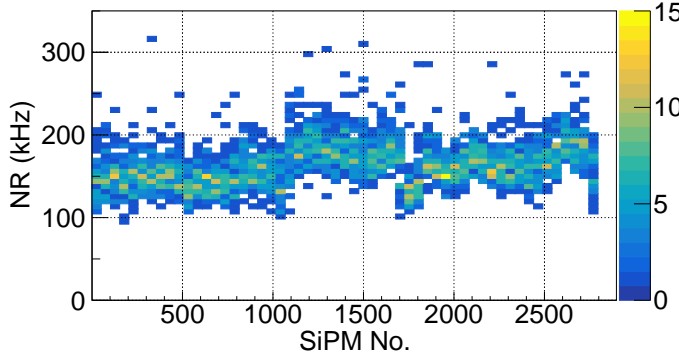

Figure 7: Noise rate (at 0.5 pe threshold) versus SiPM number.

Fig. 8a and Fig. 8b shows the correlation for dG/dV and $V_{br}$ with the noise rate respectively. No correlation is observed for these parameters with the noise rate.

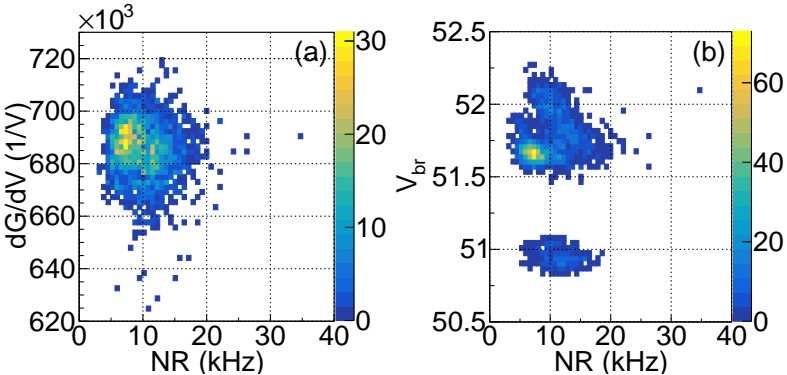

Figure 8: Correlation between (a) dG/dV and the noise rate (at 1.5 pe threshold) and (b) $V_{br}$ and the noise rate (at 1.5 pe threshold).

# 6 Conclusion

All the SiPMs required for the CMVD installation have been tested using LED calibration method. The $V_{br}$ varies from 50.7 V to 52.3 V and matches with the specification given by Hamamatsu, at 25 ° C. The spread in dG/dV is $1.2 \times 10^4 (1/V)$. The overall spread in noise rate at 0.5 p.e. threshold is 26.8 kHz for $V_{ov}$=3 V. Out of a total of 3488 SiPMs, there are only two exceptions i.e. one SiPM with low gain and one SiPM with physical damage. All other SiPMs satisfy the gain requirement as well as the noise tolerance level for the CMVD operation.

# Acknowledgements

We would like to thank Darshna Gonji, Santosh Chawan and Vishal Asgolkar for providing help for the experimental setup.

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
