# Peer review of "Mass testing of SiPMs for the CMVD at IICHEP"

_SciPost Physics Proceedings, doi:SciPost Phys. Proc. 13, 025 (2023)_

## Round 1 · Referee Report · Anonymous (Referee 1) · 2022-9-29

Strengths

The paper is well written and provides detailed description of the experimental procedure.

Weaknesses

1 - Some figures may need minor cosmetic changes such as alignments of sub-figures etc. 2. Few terms used need clear definitions in the text.

Report

The submission can be accepted for publication with suggested corrections.

Requested changes

  1. Figure 1: Labeling of mini-ICAL and CMVD detector as well as other components will improve clarity.

  2. Sec 3 - Experimental Setup: Was the setup temperature controlled? Or was temperature recorded to make sure that gain and noise rate comparison is for same conditions?

  3. Figure 3: Bus width (4) label should overlap the bus.

  4. Figure 3 and 4: Align figure captions.

  5. Sec 4 L1 "A typical example of LED ...": Following text might improve clarity. A typical example of SiPM output signal in response to LED excitation is shown in Fig 5a.

  6. Figure 5b: Not clear in the plot what the number means. It seems to be the difference between peaks. If yes, width arrow can be used to indicate properly.

  7. Figure 5c: Can Y axis be defined as Gain (G)?

  8. How Vbr is defined from Figure 5c? Add the text.

  9. Following on previous comment (7), definition of dG/dV is not immediately clear. Describing y axis of figure 5c as Gain can improve that. Similarly, equation 2 and subsequent text use $\mu$ for gain which is not consistent with dG/dV .

  10. Figure 6: Add more details to the caption.

  11. Can you plot histograms for Figure 6a, Figure 6b and Figure 7 and discuss the spread quantitatively in the conclusion?

---

## Round 2 · Referee Report · Anonymous (Referee 1) · 2022-10-17

Report

Thank you for the updated manuscript. I have made couple of suggestions and the manuscript can be accepted after implementing those minor changes.

Requested changes

1) Can you highlight that all your measurement were done at a constant temperature/room temperature? You do mention it in the caption of Fig. 6(b); however, it would be useful to highlight this earlier as well.

2) You have used wording 'Vth' in the conclusion to indicate breakdown voltage as well as over-voltage? Use consistent terminology and define Vth before using.

---

## Round 2 · List of Changes

1. Figure 1: Labeling of mini-ICAL and CMVD detector has been improved.
  2. Figure 3: Bus width (4) label should overlap the bus
  3. The label is overlapped in new version.
  4. Figure 3 and 4: Align figure captions.
  5. The captions are aligned as suggested.
  6. Sec 4 L1 "A typical example of LED ...": Following text might improve clarity. A typical example of SiPM output signal in response to LED excitation is shown in Fig 5a.
  7. Replaced as suggested.
  8. Figure 5b: Not clear in the plot what the number means. It seems to be the difference between peaks. If yes, width arrow can be used to indicate properly.
  9. Yes, it is the gap between two consecutive peaks. I have mentioned it in the text now.
  10. Figure 5c: Can Y axis be defined as Gain (G)?
  11. Yes, it can be, with a conversion factor. I have implemented the changes.
  12. How Vbr is defined from Figure 5c? Add the text.
  13. Added in the text.
  14. Following on previous comment (7), definition of dG/dV is not immediately clear. Describing y axis of figure 5c as Gain can improve that. Similarly, equation 2 and subsequent text use $\mu$ for gain which is not consistent with dG/dV .
  15. The y-axis of figure 5c is described as gain in the paper now. The text has been added to make the definitions clear.
  16. Figure 6: Add more details to the caption.
  17. I have added little more details in the captions.
  18. Can you plot histograms for Figure 6a, Figure 6b and Figure 7 and discuss the spread quantitatively in the conclusion?
  19. The spread in the numbers are added in the conclusion.

---

## Round 3 · Referee Report · Anonymous (Referee 1) · 2022-10-25

Report

The author has implemented all suggestions in the resubmitted manuscript. The manuscript may be accepted for publication.

---

## Round 3 · Author Response

Dear Sir/Madam
Thank you for the valuable comments.

Sincere regards
Mamta Jangra

---

## Round 3 · List of Changes

1) Can you highlight that all your measurement were done at a constant temperature/room temperature? You do mention it in the caption of Fig. 6(b); however, it would be useful to highlight this earlier as well.
- The following suggestion has been implemented.
2) You have used wording 'Vth' in the conclusion to indicate breakdown voltage as well as over-voltage? Use consistent terminology and define Vth before using.
- Corrected as suggested.

---

## Editorial Decision

published